# The Protective Effects of Pre- and Post-Administration of Micronized Palmitoylethanolamide Formulation on Postoperative Pain in Rats

**DOI:** 10.3390/ijms21207700

**Published:** 2020-10-18

**Authors:** Rosalba Siracusa, Roberta Fusco, Marika Cordaro, Alessio F. Peritore, Ramona D’Amico, Enrico Gugliandolo, Rosalia Crupi, Tiziana Genovese, Maurizio Evangelista, Rosanna Di Paola, Salvatore Cuzzocrea, Daniela Impellizzeri

**Affiliations:** 1Department of Chemical, Biological, Pharmaceutical and Environmental Sciences, University of Messina, Viale Ferdinando Stagno D’Alcontres, 31, 98166 Messina, Italy; rsiracusa@unime.it (R.S.); rfusco@unime.it (R.F.); aperitore@unime.it (A.F.P.); rdamico@unime.it (R.D.); egugliandolo@unime.it (E.G.); tgenovese@unime.it (T.G.); dimpellizzeri@unime.it (D.I.); 2Department of Biomedical, Dental and Morphological and Functional Imaging University of Messina, Via Consolare Valeria, 98125 Messina, Italy; cordarom@unime.it; 3Department of Veterinary Sciences, University of Messina, 98168 Messina, Italy; rcrupi@unime.it; 4Institute of Anaesthesiology and Reanimation, Catholic University of the Sacred Heart, 00168 Rome, Italy; maurizio.evangelista@unicatt.it; 5Department of Pharmacological and Physiological Science, Saint Louis University School of Medicine, Saint Louis, MO 63104 , USA

**Keywords:** palmitoylethanolamide, inflammation, pain, biochemical and therapeutic advances

## Abstract

Background: Postoperative pain (PO) is a common form of acute pain. Inadequate PO treatment is an important health problem, as it leads to worse outcomes, such as chronic post-surgical pain. Therefore, it is necessary to acquire new knowledge on PO mechanisms to develop therapeutic options with greater efficacy than those available today and to lower the risk of adverse effects. For this reason, we evaluated the ability of micronized palmitoylethanolamide (PEA-m) to resolve the pain and inflammatory processes activated after incision of the hind paw in an animal model of PO. Methods: The animals were subjected to surgical paw incision and randomized into different groups. PEA-m was administered orally at 10 mg/kg at different time points before or after incision. Results: Our research demonstrated that the pre- and post-treatment with PEA-m reduced the activation of mast cells at the incision site and the expression of its algogenic mediator nerve growth factor (NGF) in the lumbar spinal cord. Furthermore, again at the spinal level, it was able to decrease the activation of phospho-extracellular signal-regulated kinases (p-ERK), ionized calcium binding adaptor molecule 1 (Iba1), glial fibrillary acidic protein (GFAP), and the expression of brain-derived neurotrophic factor (BDNF). PEA-m also reduced the nuclear factor kappa-light-chain-enhancer of activated B cells (NF-κB) spinal pathway, showing a protective effect in a rat model of PO. Conclusion: The results obtained reinforce the idea that PEA-m may be a potential treatment for the control of pain and inflammatory processes associated with PO. In addition, pre- and post-treatment with PEA-m is more effective than treatment alone after the surgery and this limits the time of taking the compound and the abuse of analgesics.

## 1. Introduction

Pain is considered a negative experience associated with tissue damage. The distinction between acute and chronic pain remains the most common and generally determines the choice of drug therapy [1]. One of the most common causes of acute pain is postoperative pain [2,3]. A large number of surgical procedures take place every year throughout the world [4]. Such surgical interventions lead to a series of consequences including high mortality and morbidity rates that reach 10 and 16%, respectively; high costs of treatment of chronic pain resulting from acute pain; and, consequently, economic losses to society [5,6]. Therefore, in recent years, the treatment of acute pain, immediately after surgery, has gained enormous importance in the health field [7,8]. Although pain medication has made great progress, data from around the world suggest that postoperative pain continues to be inadequately managed [4,9]. The lack of clinically significant progress is probably due to the fact that the research of postoperative analgesia is often complicated by the limited efficacy and unwanted side effects of analgesic drugs currently available [10,11,12]. Surgery is known to cause the cellular and vascular release of pro-nociceptive and pro-inflammatory substances that mediate postoperative pain [13]. Over the past decade, a number of studies have been conducted to evaluate the role of the endocannabinoid system in nociceptive processing and its potential target for inducing analgesia, especially because cannabinoid receptors and ligands are found at almost all levels of pain from peripheral to central sites [14]. However, the clinical use of the agonists of this system is still limited due to its side effects and psychotropic activity. The chemical similarity of palmitoylethanolamide (PEA) with endocannabinoid anandamide [15] and the evidence on the interactions between PEA and the endocannabinoid system [16,17,18] favored the realization of a series of pharmacological studies whose results confirmed the efficacy of PEA in the promotion of inflammation resolution processes with consequent pain reduction in experimental conditions characterized by a high involvement of mast cells [19]. 

In 1996, Mazzari et al. reported for the first time the anti-inflammatory effects of PEA in an in vivo study in which it was shown that oral administration of PEA is capable of reducing mast cell degranulation and plasma extravasation induced by the injection of substance P in the mouse ear pinna [19]. The oral administration of PEA is also capable of reducing the paw edema induced by carrageenan, dextran, and formalin. These studies suggest that PEA can modulate the activation of mast cells and suppress the inflammatory responses that are activated following the degranulation of mast cells [19]. Subsequently, the anti-inflammatory effects of PEA were studied in other inflammatory diseases. In particular, the protective action of PEA has been observed in several in vivo models of inflammatory bowel diseases, such as chronic croton oil-induced colitis, dextran sodium sulphate (DSS)-induced ulcerative colitis, oil of mustard (OM)-induced accelerated transit, and dinitrobenzene sulfonic acid (DNBS)-induced colitis [20,21,22,23]. Other experimental in vitro and in vivo models by which the anti-inflammatory effect of PEA has been studied are allergic contact dermatitis (CAD), uveitis, retinal inflammation, and cystitis [24,25,26,27,28,29].

The anti-inflammatory and pain-relieving effects of PEA have also been described in conditions characterized by neurogenic inflammation and pelvic pain [20,22,30,31,32]. It is important to underline that, unlike anti-inflammatory drugs, PEA preserves the analgesic and anti-inflammatory effect even in chronic and/or neuropathic pain [33,34,35,36,37,38,39]. In addition to peripheral processes, PEA is able to contain neuroinflammation processes involving the central nervous system (CNS) in cases of neurodegenerative, psychiatric, and trauma diseases. 

Importantly, an analysis was performed to evaluate the efficacy and safety of the micronized formulation of PEA (PEA-m) on pain intensity in patients with chronic and/or neuropathic pain [40]. This study confirmed that PEA has anti-inflammatory, anti-hyperalgesia, and anti-allodynic activity. It also demonstrated that PEA administration lacked acute and chronic toxicity and was not associated with gastric mucosal lesions [40]. 

Therefore, the pharmacological characteristics of PEA and the results demonstrated in various experimental models have stimulated our research on the evaluation of PEA in its micronized form (PEA-m) in an animal model of acute postoperative pain in order to demonstrate the ability of this compound to resolve the pain and inflammatory processes activated after a surgical incision. 

## 2. Results

### 2.1. Treatment with PEA-m Relieved Mechanical Allodynia, Thermal Hyperalgesia, and Motor Coordination in Rat After Hind Paw Incision

Before incision of the leg, we observed that the rats lifted the hind paw when the pressure applied with the plastic tip reached almost 60–70 g. However, after surgery, the paw withdrawal threshold (PWT) markedly decreased to a value of 20–30 g in the postoperative pain (PO) + vehicle group, the PO + PEA-m pre + post-treatment group, and the PO + PEA-m post-treatment group. The decrease began 2 h after the incision of the paw and remained more or less stable for up to 4 h. At the sixth hour, we observed a slight increase in PWT in all three groups, even if those treated showed higher values than the vehicle. The most significant result was found at 24 h, as we observed a significant increase in PWT in the PO + PEA-m pre + post-treatment group compared to the other two groups. In fact, the animals of the treated group both before and after the incision lifted the paw after a pressure of 70 g, while the rats of the other two groups raised their paws after a pressure equal to 40 g (Figure 1A). Hind paw incision led to a time-dependent development of thermal hyperalgesia that peaked at 4 h. Treatment with PEA-m after surgery failed to show a significant reduction, although there was a trend. In contrast, PEA-m as pre + post-treatment produced a clear and significant inhibition of the development of incision-induced thermal hyperalgesia already at 6 h, and it was maintained up to 24 h (Figure 1B). As regards motor function, the animals were subjected to the Rotarod test at 2, 4, 6, and 24 h after hind paw incision. After the damage to the paw, the rats showed a series of impairments in locomotor coordination compared to time zero. The only group of animals that showed limited alteration of the locomotor function was that treated with PEA-m both before and after the paw incision (Figure 1C). 

### 2.2. Effect of PEA-m on Mast Cells (MC) infiltration and NGF Levels

Mast cells play a key role in the inflammatory process and in particular in the development of hyperalgesia. Therefore, we investigated whether treatment with PEA-m had effects on the number of mast cells by toluidine blue staining. As shown in Figure 2B and the relative graph in Figure 2E, 24 h after the posterior incision of the paw, there was a significant increase in the number of mast cells, compared with the non-injured paw of the sham group (Figure 2A and the relative graph in Figure 2E). Treatment with PEA-m after injury induction did not have a significant effect on mast cell degranulation (Figure 2C and the relative graph in Figure 2E). Instead, PEA-m as pre + post-treatment significantly reduced the presence of mast cells compared to the PO group, as shown in Figure 2D and the relative graph in Figure 2E.

Nerve growth factor (NGF) is one of the main factors that stimulate the migration, maturation, proliferation, and activation of mast cells [41]. Therefore, by immunohistochemistry analysis, we evaluated NGF levels in the lumbar area of the spinal cord. An increase in NGF was found in rats subjected to hind paw incision (Figure 3B and the relative graph in Figure 3E). We did not detect a significant reduction in animals of the PO + PEA-m post-treatment group (Figure 3D and the relative graph in Figure 3E). Instead, PEA-m pre + post-treatment significantly reduced the levels of this protein (Figure 3C and the relative graph in Figure 3E). 

### 2.3. Effect of PEA-m on p-ERK Levels

In sham-operated rats, low levels of phospho-extracellular signal-regulated kinase (p-ERK) expression were identified in the lumbar spinal cord (Figure 4A,E). However, 24 h following hind paw incision, increased p-ERK immunoreactivity was detected in the PO group (Figure 4B,E). PEA-m administered after paw incision had no effect on p-ERK expression (Figure 4C,E), whereas PEA-m as pre + post-treatment was able to significantly reduce the levels of this protein (Figure 4D,E).

### 2.4. Effect of PEA-m on Expression Levels of BDNF, Iba1, and GFAP

Immunofluorescence evaluation showed a basal expression of brain-derived neurotrophic factor (BDNF) in lumbar spinal cord section from sham animals (Figure 5A,E). Twenty-four hours after plantar incision, we found a significantly increase in BDNF-positive cells in the lumbar spinal cord from the PO group, as shown in Figure 5B and the relative graph in Figure 5E. PEA-m as pre + post-treatment was able to significantly decrease the expression of BDNF after plantar injury (Figure 5D,E). Instead, treatment with PEA-m after incision did not have significant efficacy on BDNF levels (Figure 5C,E). 

The expression levels of ionized calcium binding adaptor molecule 1 (Iba1), which is a microglial marker, and glial fibrillary acidic protein (GFAP), which is an astrocyte marker, were detected by Western blot analysis. As shown in Figure 6A,A1,B,B1, the expression levels of Iba1 and GFAP were significantly increased in the PO group, as compared with the unoperated rats group. However, PEA-m as pre + post-treatment significantly decreased the levels of Iba1 and GFAP, as compared with the operated rat group and PO + PEA-m post-treatment. These findings suggested that PEA-m administration decreased the activation of microglia and astrocytes in the rat models of inflammatory pain.

### 2.5. Effect of PEA-m on NF-κB and iNOS

To characterize the inflammatory state and possible pathways involved in PO, we used immunohistochemical staining to analyze the levels of key inflammatory mediators in the lumbar spinal cord. Our results show that the plantar incision induced an increase of nuclear factor kappa-light-chain-enhancer of activated B cells (NF-κB) in operated rats (Figure 7B,E). A reduction of NF-κB was observed in the PO + PEA-m post-treatment group, but it was not significant compared to the PO group (Figure 7C,E). PEA-m as pre + post-treatment had markedly reduced levels of this protein (Figure 7D,E).

In addition, plantar incision resulted in a pronounced increase in inducible nitric oxide synthase (iNOS) levels compared to sham rats (Figure 8B,E). PEA-m post-treatment slightly reduced the levels of this inflammatory mediator (Figure 8C,E), while PEA-m administration both before and after plantar incision caused a significant decrease in levels of iNOS (Figure 8D,E). 

## 3. Discussion

Postoperative pain is a form of acute pain. Despite the evolution in understanding neurobiology and molecular biology of postoperative pain, there are still no safe and optimal drug therapies [42]. During surgery, the tissues and nerve endings are traumatized, and this triggers an inflammatory response that causes pain at the surgical incision site [43]. Various inflammatory mediators induce peripheral and central sensitization, with a consequent increase in pain [44]. Previous studies have shown that the exogenous administration of PEA plays a key role in promoting inflammation resolution processes, resulting in pain reduction. For example, our previous studies showed that PEA-m has a protective effect on the time course of carrageenan-induced thermal hyperalgesia [45] and on glial activation and trigeminal hypersensitivity [46,47,48]. PEA-m, unlike the current drugs used for postoperative pain, does not show any side effects [46,49,50,51]. Therefore, we undertook this study to evaluate the effect of PEA-m in the rat model of postoperative pain. Patients undergoing surgery have both mechanical and thermal hyperalgesia [52,53]. The model used in this study induces well-characterized behaviors and mimics human conditions [54,55]. Therefore, we tested the effect of PEA-m in mechanical hyperalgesia using the von Frey test and in thermal hyperalgesia using the plantar test on rats. Our results showed that mechanical and thermal hyperalgesia was present on the affected limb within 2 h and persisted for 24 h after incision in the PO group. Previous studies have shown that incision-induced mechanical and thermal hyperalgesia was greater on the day of the incision [56]. For this reason, we chose to terminate the experiment 24 h after incision. Primary afferent activation and peripheral sensitization were profound immediately after surgery, and most changes in protein expression occurred when the pain behavior was more pronounced [56,57,58]. Therefore, we examined neurobiological changes in the spinal cord 24 h after surgery. PEA-m given before and after incision induced a significant reduction in mechanical hyperalgesia in rats operated 24 h after the incision, while thermal hyperalgesia was already significantly reduced 6 h after incision of the hind paw. Moreover, post-treatment with PEA-m had an impact on mechanical and thermal hyperalgesia, with a maximal lower effect compared to the group treated both before and after incision. Analogous to humans, in rats, mechanical sensitivity requires a significantly higher resolution time than heat hypersensitivity [59,60]. Another behavioral test we performed to evaluate the motor deficit after the incision was the Rotarod test. Our results showed that pre- and post-treatment with PEA-m had a better effect on the motor recovery of operated animals compared to post-treatment alone. 

Postoperative pain mechanisms involve activation, modulation, and modification at the peripheral, spinal, and cerebral levels. The pain is picked up at a peripheral level by a particular type of receptor, the so-called nociceptors, which then transmit the signal through two kinds of nerve fibers (Aδ and C). These fibers travel from the peripheral receptor (located on the skin, on a mucosa, on a serosa, or on the capsule of an organ) to the spinal cord, where they take synapses with a medullary neuron. The neuron will then transmit with its axon, through the spino-thalamic bundle, the pain message to one of the encephalic structures delegated to the elaboration of the response (cerebral cortex, thalamus, hypothalamus, etc.) [61,62]. It is known that the incision site has several signs of inflammation [63]. These processes are characterized by cellular and vascular events that result in the release of inflammatory mediators at the peripheral level [64,65]. Among these, we have the mast cells that, as seen in previous studies, are involved in the production of postoperative pain as they degranulate in the incised skin [66]. Previous studies show that PEA works by reducing the mastocyte hyperactivation that underlies very different inflammatory conditions [67,68]. Therefore, we evaluated the effect of PEA-m on mast cell degranulation at the site of incision. Our results showed that animals treated only after the operation showed a number of degranulated mast cells very similar to the PO group, while the pre- and post-treatment with PEA-m significantly limited the degranulation of mast cells at the site adjacent to the wound compared to the PO group. Peripheral inflammation can lead to central changes, particularly in the spinal cord, where secondary hyper-excitability increases and activates the neuroglial cells, which in turn release neuroinflammatory and cellular regulating factors [69,70]. Therefore, after evaluating the activation of inflammatory processes at the site of damage, we moved to the lumbar spinal cord to evaluate other inflammatory factors. It is known that mast cells release NGF [71], contributing to nociceptive signaling, hyperalgesia, and pain after incision. The proximity of the mast cell nerves in the tissue facilitates neuro-immune crosstalk relevant to pain modulation [72]. Our results showed that PEA-m administered before and after the operation is able to reach the lumbar area of the spinal cord and reduce the levels of NGF that increase following the incision of the paw. In addition to NGF, the posterior paw incision induces ERK activation at the spinal level [73]. Several studies have identified that ERK in the spinal cord contributes to the induction and development of pain. In particular, p-ERK appears to be involved in mechanical hyperalgesia [74]. Therefore, in our study, we investigated whether the compound PEA-m was able to regulate p-ERK levels in the lumbar spinal cord. Our results showed that pre- and post-treatment limited the activation of p-ERK compared to post-treatment after incision of the hind paw. In the post-operative pain model, BDNF overexpression was observed both in dorsal root neurons and in spinal cord neurons. Furthermore, the activation of microglia together with the increase of BDNF levels in the spinal cord seems to be fundamental for nociceptive signaling [75]. In this regard, we demonstrated that the expression of BDNF and Iba1 increased in the lumbar spinal cord and that pre- and post-treatment with PEA-m reduced levels of both the neurotrophic factor and the microglia marker. In addition, we saw that astrocytic levels also increased in the lumbar area of the spinal cord following the incision of the hind paw. Additionally, in this case, the double treatment with PEA-m (pre + post) reduced the number of activated astrocytes. 

It is probable that hypersensitivity to pain following peripheral inflammation is mediated by activation of the spinal pathway NF-κB [76]. From the results obtained, 24 h after the incision of the hind paw, a significant increase in NF-κB was observed in the lumbar spinal cord of the PO group. An important event in the entire series of changes is represented by the activation of the NF-κB pathway, as well as increased levels of proteins such as iNOS. The animals in which PEA-m was administered both before and after the operation had a significant reduction in the levels of both proteins. 

In conclusion, given the results obtained and given the favorable side effect profile of this compound, we can say that PEA-m could be an attractive alternative in the control of postoperative pain.

## 4. Materials and Methods

### 4.1. Animals

Adult male Sprague Dawley rats (200–250 g Envigo Italy) were housed in a controlled environment with free access to typical rodent diet and water. This study was approved by the University of Messina Review Board for the care of animals. Animal care conformed to Italian regulations on the use of animals for experimental and scientific purposes (D.Lgs 2014/26 and EU Directive 2010/63).

### 4.2. Drug

PEA was subjected to the air-jet milling technique, in which a coarse powder is slowly fed into a jet-mill apparatus endowed with a chamber of 300 mm in diameter that operates with “spiral technology” driven by compressed air. The high number of collisions that occur between particles as a result of the high level of kinetic—not mechanical—energy produces micron- and sub-micron-sized crystals [77]. PEA-m was suspended in 1% carboxymethylcellulose. The drug was administered orally via gavage at a dose of 10 mg/kg of body weight.

### 4.3. Postoperative Pain Model (PO)

According to the procedure described by Brennan et al., the model was performed on rats of postoperative pain [2]. Briefly, the animals were placed dorsally and anesthetized. After the aseptic preparation of the left posterior leg, an incision of 1 cm was made in the plantar area of the foot, starting from 0.5 cm from the back edge of the heel and continuing towards the toes of the foot. The underlying muscle was raised with curved forceps and longitudinally incised, leaving the origin and insertion of the muscle intact, and the skin was then closed with two interrupted mattress sutures using a 5-0 nylon suture. The rats in the control group received anesthesia but did not receive an incision. 

### 4.4. Experimental Groups

Sham + vehicle: rats in the control group received anesthesia but did not receive an incision. Vehicle solution (1% carboxymethylcellulose and saline) was administrated orally. (*N* = 10).Sham + PEA-m: rats in the control group received anesthesia but did not receive an incision. PEA-m was administrated orally 3 days before hind paw incision and 1, 6, and 8 h after surgery. (*N* = 10) (data not shown).PO + vehicle: after anesthesia, a longitudinal incision was performed on the right hind paw, and the vehicle (1% carboxymethylcellulose and saline) was then administered orally. (*N* = 10).PO + PEA-m post-treatment: the same as the PO + vehicle, but rats were treated with PEA-m (10 mg/kg oral somministration) 1, 6, and 8 h after hind paw incision. (*N* = 10).PO + PEA-m pre + post-treatment: the same as the PO + vehicle group, but rats were treated with PEA-m (10 mg/kg oral somministration) 3 days before hind paw incision and 1, 6, and 8 h after surgery. (*N* = 10).

The dosage of PEA-m was selected on the basis of previous in vivo studies [45,78].

### 4.5. Behavioral Analysis

#### 4.5.1. Mechanical Hyperalgesia

In animals, hypersensitivity to a mechanical stimulus was evaluated by the electronic von Frey filament test (Bioseb, Chaville, France) consisting of a portable force transducer equipped with a plastic tip [79]. Briefly, rats were placed in plastic boxes, under which there was a metal mesh floor, and were made to acclimatize for 15 min before starting the test. Thereafter, the transducer tip was placed perpendicular to the medial plantar surface of the hind paw until a sharp lift was observed. The mechanical threshold (expressed in grams) that corresponds to the pressure that stimulates a behavioral reaction (withdrawal of the hind paw) was recorded automatically by the electronic device. Generally, for each animal, 2 measurements were made at intervals of at least 3 min, and an average of the obtained values was then made.

#### 4.5.2. Thermal Hyperalgesia

Thermal hyperalgesia was evaluated using the plantar test (Hargreaves method, Ugo Basile), which allows one to measure the latency of the withdrawal of the paw following a thermal stimulus [80]. The apparatus used consists of a mobile unit containing a heat source that emits a beam of light. Briefly, the rats were placed in Plexiglas chambers and allowed to habituate. The mobile unit was then placed under a single hind paw in order to provide the thermal stimulus. At this point, the paw latency period was determined with an electronic clock circuit and thermocouple. Measurements were taken in duplicate at least 1 min apart, and the average was used for statistical analysis. The results obtained are expressed as paw withdrawal latencies.

#### 4.5.3. Rotarod Test

Motor coordination was evaluated using the rotarod test [81]. Twenty-four hours before testing, we submitted all animals to a training session until they could remain in the apparatus for 60 s without falling. Each training was performed at the minimal speed for training sessions of 1–2 min at intervals of 30–60 min. After this learning period, the rats were placed onto the rotarod at a constant speed of 25 rpm. Subsequently, the accelerator mode was selected on the treadmill, i.e., the rotation rate of the drum was increased linearly at 20 rpm. The time was measured from the start of the acceleration period until the rat fell off the drum. The cutoff time was 30 s. During the test session, the latency (s) for the first fall and the total number of falls over a 4 min period were observed.

### 4.6. Toluidine Blue Staining

Skin sections were deparaffinized in xylene and dehydrated by a graded succession of ethanol, 5 min in each solution [82]. The sections were next placed in water for 5 min, relocated to toluidine blue for 4 min, and then blotted cautiously. Sections were positioned in absolute alcohol for 1 min, cleared in xylene, and fixed on glass slides using Eukitt (Bio-Optica, Milan, Italy). The number of metachromatic stained mast cells was obtained by counting 5 high-power fields (40×) per section using a DM6 microscope (Leica, Milan, Italy).

### 4.7. Immunostaining of NGF, p-ERK, NF-kB, and iNOS

Briefly, at 24 h after surgery, the lumbar spinal cord tissues were fixed in 10% buffered formaldehyde and 7 μm sections were prepared from paraffin-embedded tissues. After deparaffinization, endogenous peroxidase was quenched with 0.3% H_2_O_2_ in 60% methanol for 30 min. The sections were permeabilized with 0.1% Triton X-100 in Phosphate Buffered Saline (PBS) for 20 min. Non-specific adsorption was minimized by incubating the section in 2% normal goat serum in PBS for 20 min. Endogenous biotin or avidin binding sites were blocked by sequential incubation for 15 min with avidin and biotin [83]. Slides were incubated overnight with either an anti-NGF mouse monoclonal antibody (Santa Cruz Biotechnology, CA, USA, 1:250 in PBS, *v*/*v*), an anti-p-ERK mouse monoclonal antibody (Santa Cruz Biotechnology, CA, USA, 1:250 in PBS, *v*/*v*), an anti-NF-kB mouse monoclonal antibody (Santa Cruz Biotechnology, CA, USA, 1:250 in PBS, *v*/*v*), and an anti-iNOS mouse monoclonal antibody (Santa Cruz Biotechnology, CA, USA, 1:250 in PBS, *v*/*v*). Slides were then washed with PBS and incubated with a secondary antibody. Specific labeling was identified with an avidin-biotin peroxidase complex and a biotin-conjugated goat anti-rabbit Immunoglobulin G (IgG) (Vector Labs Inc., Burlingame, CA). To verify antibody-binding specificity, we also incubated some slides with only a primary antibody or secondary antibody; no positive staining was found. Immunohistochemical images were evaluated by densitometric analysis using an imaging densitometer (DM6, Leica, Milan, Italy).

### 4.8. Immunofluorescence for BDNF

After deparaffinization and rehydration, the detection of BDNF was carried out after boiling sections in 0.1 M citrate buffer for 1 min. Non-specific adsorption was minimalized by incubating in 2% (*v/v*) standard goat serum in PBS for 20 min. Lumbar spinal cord sections were incubated overnight with murine monoclonal anti-BDNF antibodies (1:100, Santa Cruz Biotechnology, Santa Cruz, CA, USA) at 37 °C in a humidified oxygen and nitrogen chamber. Sections were then incubated with a secondary antibody—a Fluorescein isothiocyanate (FITC) conjugated anti-mouse Alexa Fluor-488 antibody (1:2000 *v*/*v* Molecular Probes, United Kingdom)—for 1 h at 37 °C. Nuclei were stained by adding 2 μg/mL 4′,6′-diamidino-2-phenylindole (DAPI; Hoechst, Frankfurt, Germany) in PBS. Sections were observed at 20× magnifications by a Leica DM2000 microscope (Leica, Milan, Italy). Optical sections of samples were obtained by an HeNe laser (543 nm), a UV laser (361 to 365 nm), and an argon laser (458 nm) at a 1 min, 2 s scanning rapidity with up to 8 averages; 1.5 μm sections were attained using a pinhole of 250. Examining the most luminously labeled pixels and using settings that allowed clear visualization of structural details, while keeping the maximum pixel intensities close to 200, established contrast and brightness. The same settings were used for all images obtained from the other samples that had been processed in parallel. Digital images were cropped and figure montages produced using Adobe Photoshop 7.0 (Adobe Systems; Palo Alto, California, United States).

### 4.9. Western Blot Analysis for GFAP and Iba1

Nuclear and cytosolic extracts were prepared as described previously [84]. Lumbar spinal cord tissue from each animal was suspended in Extraction Buffer A containing 0.2 mM phenylmethylsulfonyl fluoride (PMSF; Sigma-Aldrich), 0.15 mM pepstatin A, 20 mM leupeptin, and 1 mM sodium orthovanadate, homogenized for 2 min, and centrifuged for 4 min at 4 °C at 12,000 rpm. Supernatants represented the cytosolic fraction. The pellets, containing enriched nuclei, were resuspended in Buffer B containing 1% Triton X-100, 150 mM NaCl, 10 mM Tris HCl (pH 7.4), 1 mM ethylene glycol tetraacetic acid (EGTA; Sigma-Aldrich), 1 mM Ethylenediaminetetraacetic acid (EDTA; Sigma-Aldrich), 0.2 mM PMSF, 20 mM leupeptin, and 0.2 mM sodium orthovanadate. After they were centrifugated for 10 min at 12,000 rpm at 4 °C, the supernatants containing the nuclear protein were stored at −80 °C for further analysis. The levels of GFAP and Iba1 were measured in cytosolic fractions. The filters were blocked with 1× PBS and 5% (*w/v*) non-fat desiccated milk for 40 min at room temperature and successively probed with a primary antibody—either mouse monoclonal anti-GFAP (1:500; Santa Cruz Biotechnology, CA, USA) or mouse monoclonal anti-Iba1 (1:500; Santa Cruz Biotechnology, CA, USA)—at 4 °C overnight in 1× PBS, 5% (*w/v*), non-fat dried milk, and 0.1% Tween-20. Membranes were incubated with peroxidase-conjugated goat anti-rabbit IgG or peroxidase-conjugated bovine anti-mouse IgG secondary antibody (1:2000, Jackson ImmunoResearch, West Grove, PA, USA) for 1 h at room temperature. To assess whether blots were loaded with equal volumes of protein lysates, we probed them with a mouse monoclonal β-actin antibody (1:5000; Santa Cruz Biotechnology, CA, USA). Signals were detected with Super Signal West Pico Chemiluminescent Substrate according to the manufacturer’s instructions (Pierce Thermo Scientific, Rockford, IL, USA). The relative expression of protein bands was quantified by densitometric scanning of the X-ray films with a GS-700 Imaging Densitometer (GS-700, Bio-Rad Laboratories, Milan, Italy) and a computer program (ImageJ), and was standardized to β-actin.

### 4.10. Statistical Evaluation

All values in the figures and text are expressed as mean ± standard error of the mean (SEM) of *N* observations. For in vivo studies, *N* represents the number of animals studied. Results displayed in the figures are representative of at minimum 3 experiments performed on diverse in vivo experimental days. The results were examined by one- or two-way analysis of variance followed by a Bonferroni post-hoc test for multiple comparisons. A *p*-value of less than 0.05 was considered significant.

## 5. Patents

Salvatore Cuzzocrea is co-inventor on patent WO2013121449 A8 (Epitech Group Spa), which deals with methods and compositions for the modulation of amidases capable of hydrolyzing N-acylethanolamines employable in the treatment of inflammatory diseases. This invention is wholly unrelated to the present study. Moreover, Prof. Cuzzocrea is also, with Epitech Group, a co-inventor on the following patent: EP 2 821 083; MI2014 A001495; 102015000067344, which is unrelated to the present study. 

## Figures and Tables

**Figure 1 ijms-21-07700-f001:**
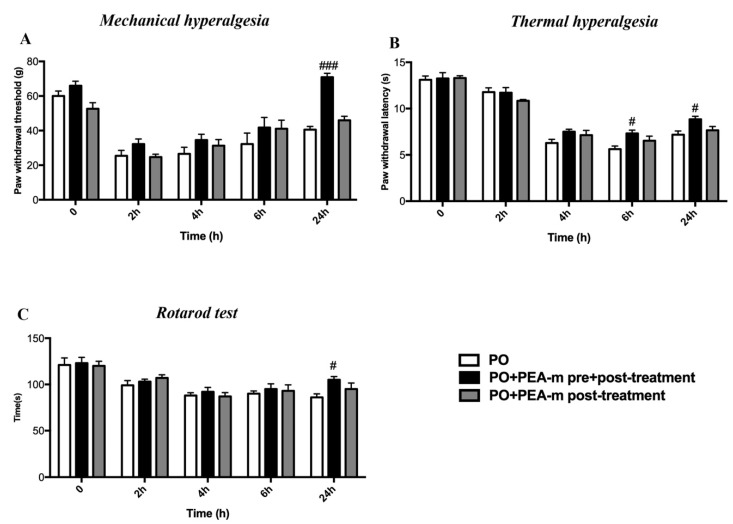
Effects of micronized palmitoylethanolamide (PEA-m) on mechanical allodynia, thermal hyperalgesia, and motor coordination after postoperative pain (PO). (**A**) Mechanical allodynia was measured using the electronic von Frey test before and after postoperative pain induction; (**B**) thermal hyperalgesia was measured using plantar test before and 2, 4, 6, and 24 h after incision; (**C**) motor deficit was measured using the Rotarod test before and 2, 4, 6, and 24 h after hind paw incision. ### *p* < 0.001 vs. PO; # *p* < 0.05 vs. PO.

**Figure 2 ijms-21-07700-f002:**
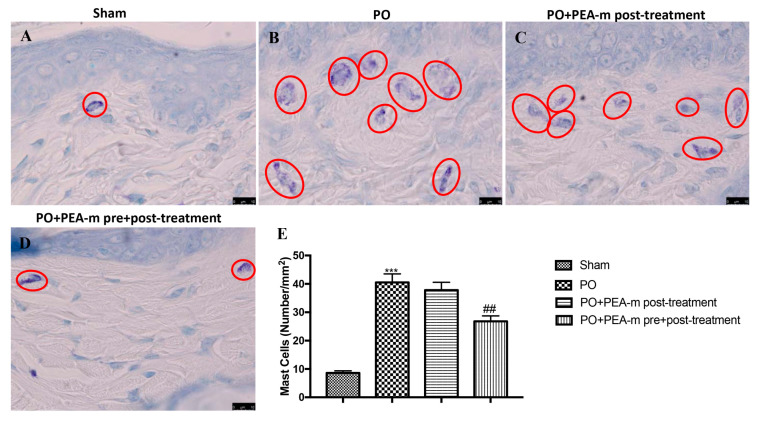
Effects of PEA-m on Mast Cells. Toluidine blue staining was used to identify mast cell infiltration (encircled in red), characterized by dark lilac blue granules: (**A**) sham group; (**B**) PO group; (**C**) PO + PEA-m post-treatment group; (**D**) PO + PEA-m pre + post-treatment group. (**E**) Mast cell number per unit area of tissue (mast cell density). Figures are illustrative of at least three distinct experiments. Values are means ± standard error of the mean (SEM) of five animals for each group. *** *p* < 0.001 vs. sham, ## *p* < 0.01 vs. PO.

**Figure 3 ijms-21-07700-f003:**
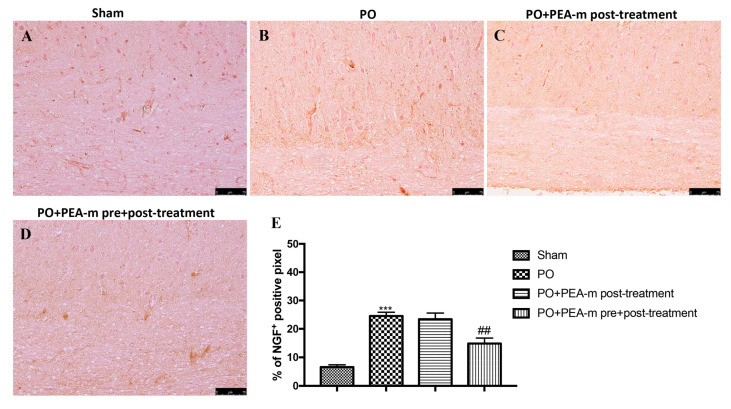
Effects of PEA-m on levels of nerve growth factor (NGF). Immunohistochemical analysis showed no staining for NGF in the sham group (**A**). Increased NGF expression was observed in lumbar spinal cord collected 24 h after PO (**B**). A non-significant reduction was observed in PO + PEA-m post-treatment group (**C**), while low levels of this protein were found in rats treated with PEA-m both before and after plantar incision (**D**). The data are expressed as percentage of pixel cells (**E**). *******
*p* < 0.001 vs. sham; ## *p* < 0.01 vs. PO.

**Figure 4 ijms-21-07700-f004:**
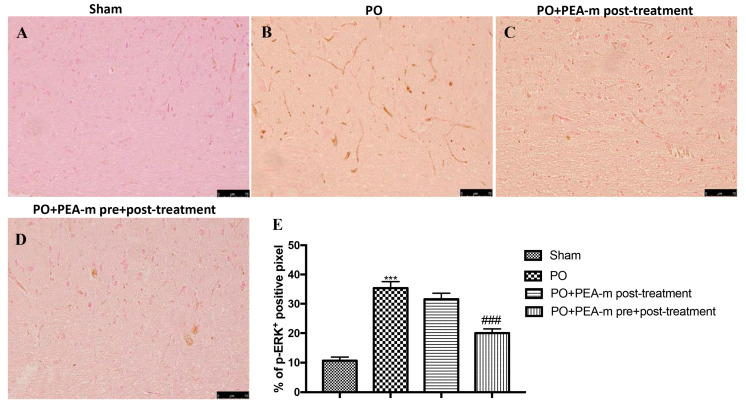
Effects of PEA-m on levels of phospho-extracellular signal-regulated kinase (p-ERK). Immunohistochemical analysis showed no staining for p-ERK in the sham group (A). Increased p-ERK expression was observed in lumbar spinal cord collected 24 h after PO (B). A non-significant reduction was observed in the PO + PEA-m post-treatment group (C), while low levels of p-ERK were found in rats treated with PEA-m both before and after plantar incision (D). The data are also presented graphically as a percentage of positive cells (E). *******
*p* < 0.001 vs. sham; ### *p* < 0.001 vs. PO.

**Figure 5 ijms-21-07700-f005:**
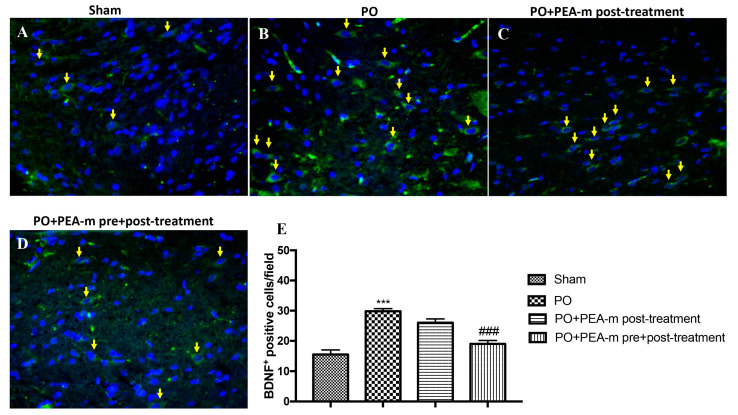
Effects of PEA-m on brain-derived neurotrophic factor (BDNF) expression in lumbar spinal cord after PO. Immunofluorescence for BDNF expression (green) in sham rats (**A**), the PO group (**B**), the PO + PEA-m post-treatment group (C), and the PO + PEA-m pre + post-treatment group (**D**). Yellow arrows show BDNF expression. Data are demonstrative of at least three separate experiments. Images are representative of all animals in each group. The graphs next to the panel represent the positive BDNF cells. All images were digitalized at a resolution of 8 bits into an array of 2048 × 2048 pixels. Pictures were captured at 40× magnification. (**E**) *** *p* < 0.001 vs. sham; ### *p* < 0.001 vs. PO.

**Figure 6 ijms-21-07700-f006:**
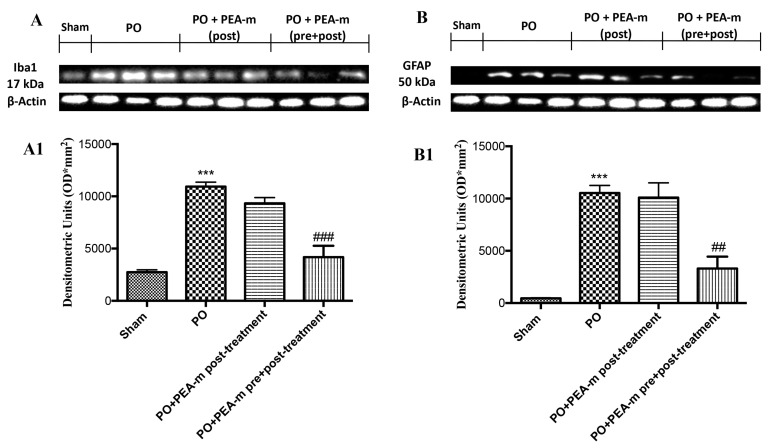
Effects of PEA-m on ionized calcium binding adaptor molecule 1 (Iba1) and glial fibrillary acidic protein (GFAP) expression after PO. Representative Western blots showing the effects of PEA-m on (**A**,**A1**) Iba1 and (**B**,**B1**) GFAP expression at 24 h after PO. PEA-m as pre + post-treatment reduced Iba1 (**A**,**A1**) and GFAP (**B**,**B1**) expression. A representative blot of lysates from five animals per group is shown, together with a densitometric analysis for all animals. The results in (A) and (B) are expressed as means ± SEM of five animals for each group. (**A1**) *** *p* < 0.001 vs. sham; ### *p* < 0.001 vs. PO; (**B1**) *** *p* < 0.001 vs. sham; ## *p* < 0.01 vs. PO.

**Figure 7 ijms-21-07700-f007:**
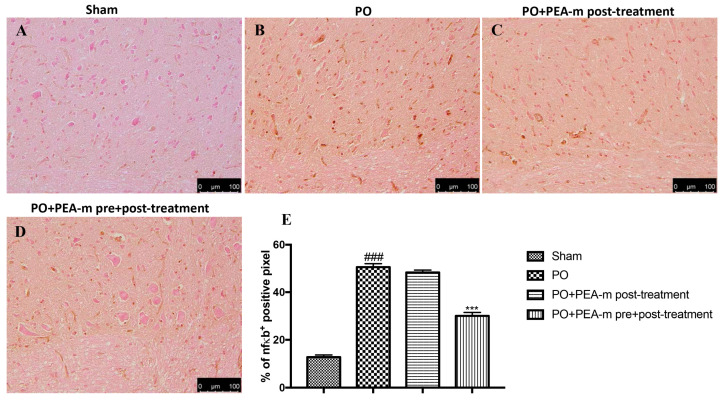
Effects of PEA-m on levels of nuclear factor kappa-light-chain-enhancer of activated B cells (NF-κB). Immunohistochemical analysis showed no staining for NF-κB in the sham group (**A**). Increased NF-κB expression was observed in the lumbar spinal cord collected 24 h after PO (**B**). A non-significant reduction was observed in the PO + PEA-m post-treatment group (**C**), while low levels of NF-κB were found in rats treated with PEA-m both before and after plantar incision (D). The data are also presented graphically as percentage of positive cells (**E**). *******
*p* < 0.001 vs. sham; ### *p* < 0.001 vs. PO.

**Figure 8 ijms-21-07700-f008:**
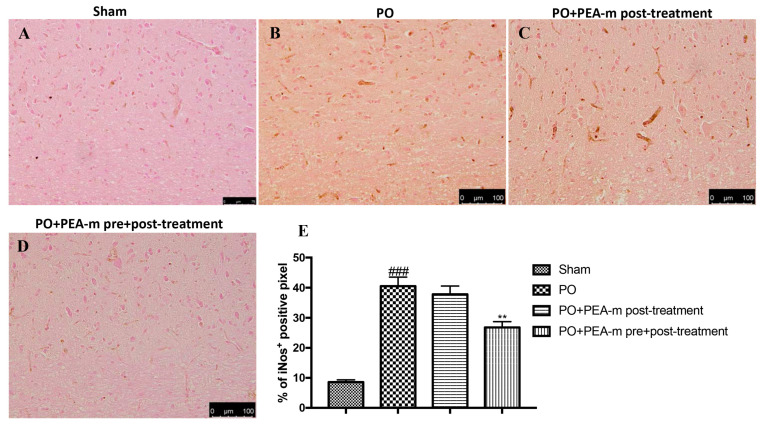
Effects of PEA-m on levels of inducible nitric oxide synthase. Immunohistochemical analysis showed no staining for iNOS in the sham group (**A**). Increased iNOS expression was observed in the lumbar spinal cord collected 24 h after PO (**B**). A non-significant reduction was observed in the PO + PEA-m post-treatment group (**C**), while low levels of iNOS were found in rats treated with PEA-m both before and after plantar incision (**D**). The data are also presented graphically as a percentage of pixel cells (**E**). **###**
*p* < 0.001 vs. sham; ** *p* < 0.01 vs. PO.

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
