# Peer review of "The Protective Effects of Pre- and Post-Administration of Micronized Palmitoylethanolamide Formulation on Postoperative Pain in Rats"

_ijms, 2020, doi:10.3390/ijms21207700_

Round 1

Reviewer 1 Report

The paper submitted for publication by Syracuse and colleagues summarizes
the results of a careful and thorough research.
The results of in vivo
pharmacological methods used in the work and related morphological and
biochemical experiments convince the reader that micronized
Palmitoylethanolamide (PEA-m) treatment before and after surgery was
effective in reducing postoperative pain in animal models.
Based on the
results presented, the oral drug may be suitable for the treatment of
one of the most common types of acute pain, postoperative pain. 
For my part, I consider the dissertation to be worth publishing in its current form and support its publication to the best of my belief.

Author Response

The paper submitted for publication by Syracuse and colleagues summarizes the results of a careful and thorough research. The results of in vivo pharmacological methods used in the work and related morphological and biochemical experiments convince the reader that micronized Palmitoylethanolamide (PEA-m) treatment before and after surgery was effective in reducing postoperative pain in animal models. Based on the results presented, the oral drug may be suitable for the treatment of one of the most common types of acute pain, postoperative pain. For my part, I consider the dissertation to be worth publishing in its current form and I support its publication to the best of my belief.

We thank the referee for the time spent reading our manuscript and for the positive comment

Reviewer 2 Report

title 

since the administration of PEA was before and after the incision and was found efficient it should be changed accordingly 

abstract 

conclusion authors should precise that PEA was efficient when administered before and after incision 

Introduction 

the inflammatory response of postoperative pain is well documented I suggest to shorten this part and expand instead properties of PEA 

FIGURE 1 columns need more contrast 

a reference about the  lack of side effects of PEA should be provided 

methodology: what was the rationale to give PEA  before and afet the incision ? some reference is needed 

Author Response

title 

since the administration of PEA was before and after the incision and was found efficient it should be changed accordingly 

As suggested by the referee, we rewrote the title to highlight that we administered PEA both before and after surgery.

abstract 

conclusion authors should precise that PEA was efficient when administered before and after incision 

As suggested by the referee, we specified in the conclusion of the abstract that PEA was efficient when administered before and after surgery 

Introduction 

the inflammatory response of postoperative pain is well documented I suggest to shorten this part and expand instead properties of PEA 

Thanks for your suggestion, we rewrote the introduction section to highlight the properties of PEA.

FIGURE 1 columns need more contrast 

As suggested by the referee, we have adjusted the contrast of the columns of figure 1

a reference about the lack of side effects of PEA should be provided 

As suggested by the referee, we added the reference about the lack of side effects of PEA.

methodology: what was the rationale to give PEA before and afet the incision? some reference is needed 

Studies in the literature show that pre-operative analgesics (administered before surgical incision) have the action of reducing central sensitization and therefore the progression to chronic pain. However, there is little evidence that demonstrates the beneficial effect of preventive administration on acute postoperative pain. Therefore, given the analgesic and anti-inflammatory properties of PEA studied in different experimental models of acute and chronic pain (D. Reddi, 2015; Linda Gabrielsson, 2016; Ian Gilron, 2019; Daniela Impellizzeri, 2014; Roberta Fusco, 2017; Daniela Impellizzeri, 2019), we wanted to evaluate whether the pre- and post-administration of the PEA-m compound was able to attenuate the physiological sequelae of nociception both locally (incision area) and at the peripheral level. Furthermore, the purpose of our study for the translationality of the results in the clinic was also to find a compound capable of reducing treatment times by limiting the abuse of analgesic compounds.

As suggested by the referee, we added the reference in the experimental design section.